# Preliminary Estimation of Nutritional Quality of the Meat, Liver, and Fat of the Indigenous Yakutian Cattle Based on Their Fatty Acid Profiles

**DOI:** 10.3390/foods12173226

**Published:** 2023-08-27

**Authors:** Olesia N. Makhutova, Vasiliy V. Nokhsorov, Kirill N. Stoyanov, Lyubov V. Dudareva, Klim A. Petrov

**Affiliations:** 1Institute of Biophysics of Federal Research Center “Krasnoyarsk Science Center” of Siberian Branch of Russian Academy of Sciences, Akademgorodok, 660036 Krasnoyarsk, Russia; makhutova@ibp.krasn.ru (O.N.M.); ikirill97@gmail.com (K.N.S.); 2School of Fundamental Biology and Biotechnology, Siberian Federal University, 79 Svobodny Pr., 660041 Krasnoyarsk, Russia; 3Institute for Biological Problems of Cryolithozone of Siberian Branch of the Russian Academy of Sciences, 41 Lenina Av., 677000 Yakutsk, Russia; kap_75@bk.ru; 4Siberian Institute of Plant Physiology and Biochemistry, Siberian Branch of Russian Academy of Sciences, 132 Lermontova Str., 664033 Irkutsk, Russia; laser@sifibr.irk.ru

**Keywords:** polyunsaturated fatty acids, lipid dietary indexes, Siberian cattle, muscle tissue, liver

## Abstract

The Yakutian cattle is an indigenous Siberian cattle breed living in an extremely cold climate in some parts of Yakutia. There are only a few thousand animals of this breed, and the conservation of the Yakutian cattle is embedded in the international agenda. We studied the fatty acid profiles in the meat, liver, and fat of the Yakutian cattle (five individuals) of different ages and their main food resource–pasture plants. The fatty acid profile of the tissues of the Yakutian cattle differed from that of pasture plants: 16:0, 18:2n–6, and 18:3n–3 dominated in the pasture plants; 16:0, 18:0, 18:1n–9, 18:2n–6, 20:4n–6, 20:5n–3, and 22:5n–3 dominated in the meat and liver; and 16:0, 18:0, and 18:1n–9 dominated in the fat. The fatty acid composition of food products is related to the risk of developing cardiovascular disease (CVD). The meat and liver of the Yakutian cattle are health food products that contribute to decreasing the risk of developing CVD because of their rather high content of eicosapentaenoic and docosahexaenoic fatty acids, optimal n–6/n–3 and polyunsaturated fatty acids/saturated fatty acids ratios, low values of indexes of atherogenicity and thrombogenicity, and high values of hypocholesterolemic/hypercholesterolemic and health-promoting indexes. The results of the present study support the importance of preserving this valuable cattle breed. Actions should be taken to increase their population while retaining their contemporary housing and feeding conditions.

## 1. Introduction

Omega–3 (n–3) long-chain polyunsaturated fatty acids (LC-PUFAs) such as eicosapentaenoic (EPA; 20:5n–3) and docosahexaenoic (DHA; 22:6n–3) acids are physiologically valuable fatty acids, which play a major role in the function of the cardiovascular and nervous systems in humans [1,2,3]. The genes involved in the synthesis of these PUFAs from α-linoleic acid (ALA, 18:3n–3) have been found in various animals, but the rate of synthesis is generally ineffective, and the requirement for PUFA is not satisfied [4,5]. The principal sources of n–3 PUFA for humans are products of aquatic ecosystems—mainly fish [6,7]. Commercial catches, however, have reached their maximum, and to increase them is to reduce fish production in the global ocean [8,9,10]. At the same time, humans suffer from n–3 PUFA deficiency. The daily consumption of EPA and DHA by an average person is 0.11 g, while the necessary amount is 1 g [7]. In the Western diets today, the proportion of the n–6 PUFA is one order of magnitude greater than the proportion of the n–3 PUFA although the appropriate n–6/n–3 ratio is 1:1–3:1 [11].

Alternative terrestrial sources of n–3 PUFA in the human diet may reduce the pressure on aquatic ecosystems. However, the n–6/n–3 ratio of terrestrial ecosystems is generally higher than that of aquatic ecosystems [12]. In addition to that, tissues of domestic animals fed grain-based artificial feeds contain higher n–6/n–3 ratios than tissues of wild animals, whose diets contain ALA-rich grasses [13,14,15]. For example, the n–6/n–3 ratio in the meat of sheep, beef cattle, horse, bison, deer, and wapiti grazed on pasture is 1–3, while in the meat of barn-fed sheep, beef cattle, pig, and bison, it is 5–9 [16,17,18,19,20,21]. This difference has been noted for most animals [22]. However, there are literature data showing high n–6/n–3 ratios even in pasture-grazed animals [23]. During the stabling period, the main forage for some farm animals is grass cut and air-sun-dried (natural drying) with a moisture content of 15–17% or lower [24]. Indeed, mowing is the primary means of forage collection in grasslands worldwide [25]. However, mowing may affect the plant quality and production [26], which causes changes in the quality of forage during the stabling period compared with the pasture period.

As a rule, in terrestrial animals, the main representatives of n–6 PUFAs are 18:2n–6 and 20:4n–6, and n–3 PUFAs are represented by 18:3n–3 rather than by EPA and DHA [12]. However, the meat of some animals such as fallow deer (*Dama dama*) was reported to contain rather high amounts of EPA + DHA: 0.7 mg/g fresh meat [27]. Thus, tissues of pasture-grazed animals generally contain optimal n–6/n–3 ratios, which alone is an indication of a healthy product. Some of the species and breeds of these animals may have meat containing, along with 18:3n–3, considerable amounts of LC-PUFA.

Extensive development of agriculture rapidly decreases biodiversity all over the world [28]. To increase production efficiency, high-yield breeds replace indigenous breeds, which are only preserved in marginal agricultural areas [29].

Indigenous animal breeds are of particular interest to researchers. They are genetically diverse and have unique traits, which are absent in the widely occurring animal breeds [29].

Yakutian cattle are the last remaining indigenous cattle breed of the East Asian “Turano-Mongolian” type of *Bos taurus* in Siberia. They are distributed in the north-eastern region of the Republic of Sakha (Yakutia) of the Russian Federation [30]. The conservation of the Yakutian cattle is embedded in the international agenda [29]. The physiological traits of Yakutian cattle make this breed a valuable source of genetic material for agriculture in subarctic regions. The Yakutian cattle have such traits as a solid torso, short strong legs, and a long thick coat. Effective thermoregulation, quick formation of subcutaneous adipose tissue, and low metabolic rates at low temperatures (even at –60 °C) allow these animals to survive in harsh environments and under poor feeding conditions [29,30,31]. Another important reason why Yakutian cattle thrive is that they consume green cryo-fodder, which accumulates higher contents of proteins, carbohydrates, and fats compared to warm-season grass [32,33,34]. 

Since 1929, the Yakutian cattle have been crossed with the Simmental and the Kholmogory cattle breeds to increase their productivity. The genetic study conducted in 2005 compared two hybrid cattle populations [30]. The study showed that the genetic contribution of the indigenous Yakutian cattle to the Yakutian–Kholmogory population was low, whereas a substantial genetic contribution of the Yakutian cattle was revealed in the genome of the Yakutian–Simmental population. Purebred Yakutian cattle can still be found in remote northern family farms in Yakutia. There is every reason to believe that the meat of indigenous Yakutian cattle, which are grazed on pasture in a harsh climate, has a distinctive composition of polyunsaturated fatty acids. Thus, the purpose of the present study was to estimate (a) the nutritional value of the liver, muscle tissue, and adipose tissue of the indigenous Yakutian cattle based on their fatty acid composition and content; (b) the fatty acid composition and content of Yakutian pasture plants (before and after mowing), which are the staple food for this cattle breed.

## 2. Materials and Methods

### 2.1. Collecting Samples of Pasture Plants Constituting the Diet of the Yakutian Cattle

The plants used in the study were various wild pasture plants growing in the north-eastern part of Yakutia (the village of Batagay-Alyta, Eveno-Bytantaysky District, the Republic of Sakha (Yakutia), Russia, 67°47′ N, 130°24′ E). They were collected in steppe meadows at three locations, with 8–10 samples from each. Samples were collected in the morning (9:00–11:00). A plant shoot pruner (Instrum-Agro, Colibri, Hong Kong, China) was used for mowing. Segments of leaves of pasture plants before and after mowing were compared (see the experimental design, Figure 1). The samples were transported in liquid nitrogen in Dewar vessels to the laboratory, where they were dried in a lyophilizer (VirTis, New York, NY, USA) and kept for further biochemical analysis.

### 2.2. The Design of the Experiment with Pasture Plants

The experiment was conducted in 2021. To assess the effect of traumatic damage of shoots (mowing) on the lipid composition of grasses, we conducted the following experiment. Freshly harvested leaves of grasses (Poaceae: *Calamagrostis neglecta*, *C. langsdorfii*, *Arctophila fulva*, *Glyceria aquatica,* and *Beckmannia syzigachne*) were used as control (Table 1). 

They were collected on 1 July and designated as “Before mowing” (Figure 1a). Then, grass shoots were cut at 4–5 cm height, and cut leaves were kept for 72 h in natural conditions and designated as “After mowing” (Figure 1b). The average air temperature during the growing season (June-September) was 13.2 °C, and the amount of precipitation from May to September was 153.2 mm. Meteorological parameters during the research period are shown in Table 2. Leaf samples (2–5 g) were taken from a mixture of different grasses without a root system from a plot of 2 m^2^ in triplicate. 

### 2.3. Analysis of Fatty Acids of Pasture Plants

The fatty acid analysis of the plant samples included lipid extraction, transesterification of fatty acids (to form FAME), and their further purification. The plant samples (≈0.5 g) were ground and homogenized in a mixture of chloroform and methanol (2:1, *v*/*v*). The reagents and laboratory glassware were cooled before use. To avoid lipid oxidation during laboratory procedures, ionol was added to the solvent mixture (1.25 mg per 100 mL of the mixture). After thorough mixing, the samples were left for 30 min until lipids were completely diffused into the solvent. Extracted lipids were purified to remove nonlipid components using a separatory funnel to which distilled water was added. The chloroform fraction containing extracted lipids was separated. Chloroform was removed from the lipid extract under vacuum using an RVO-64 rotary evaporator (Prague, Czech Republic). 

Fatty acid methyl esters (FAMEs) were produced by the method described in detail elsewhere [35]. FAMEs were purified using thin-layer chromatography on glass plates with KSK silica gel (Sorbfil, Krasnodar, Russia). Benzene was used as the mobile phase. The retention factor (Rf) of the FAME zone was 0.71–0.73. FAMEs were eluted from silica gel with hexane and analyzed by GC-MS (Model 5973/6890N, Agilent Technologies, Santa Clara, CA, USA) [36]. The content of fatty acids (mg/g of dry weight) was quantified based on the peak square of methyl nonadecanoate (Sigma-Aldrich, Burlington, MA, USA), a certain amount of which was added to the samples before the extraction of lipids.

### 2.4. Harvesting the Tissues of the Yakutian Cattle

Tissue samples were harvested in family farms that still maintain the indigenous Yakutian cattle in the fall of 2019 when the farmers were slaughtering part of their livestock for personal consumption (Eveno-Bytantaysky District, Yakutia, Russia, 68° N, 129° E). Samples of muscles (trapezius, triceps brachii, tensor fascia latae, biceps femoris, and gluteus medius), subcutaneous fat (from the rump), visceral fat (from internal organs), and liver were harvested from cattle of different ages: four males—a 7-month-old one, two 18-month-old ones, a 3-year-old one, and a 9-year-old female. During their lifetime, the cattle were mainly pasture-grazed or fed on dried grass. For the five months before slaughter, the cattle were pasture-fed. Typical pasture plants—the staple food for the cattle—were collected before and after mowing for FA composition analysis. All samples for FA analysis were kept in vials with chloroform and methanol (2:1, *v*/*v*) at −20 °C.

### 2.5. Analysis of Fatty Acids in Yakutian Cattle Tissues 

The fatty acid analysis of the animal tissue samples included lipid extraction and transesterification of fatty acids and their further purification, which were described in detail elsewhere [37]. The FAME analysis was conducted using a GC-MS (Model 7000 QQQ, Agilent Technologies, Santa Clara, CA, USA). A 30 m (internal diameter is 0.25 mm) capillary HP-FFAP column was used. Detailed descriptions of the chromatographic and mass-spectrometric conditions are given elsewhere [21]. Peaks of FAME were identified by comparing their mass spectra to those in the integrated database NIST Mass Spectral Search 2.0 (built 22 October 2009) and to mass spectra of FAME in the standard of 37 FAME mixture (U-47885, Supelco, Bellefonte, PA, USA). The FAMEs were quantified according to the peak area of the internal standard—the methyl nonadecanoate (Sigma-Aldrich, St. Louis, MO, USA)—which was added to the samples prior to the lipid extraction.

Lipid dietary indexes, such as the index of thrombogenicity (IT), index of atherogenicity (IA), health-promoting index (HPI), and hypocholesterolemic/hypercholesterolemic index (HH), were calculated using the equations described elsewhere [38,39].
IT = Ʃ (14:0, 16:0, 18:0)/Ʃ (0.5 × MUFA, 0.5 × n–6 PUFA, 3 × n–3 PUFA, n–3/n–6)(1)
IA = Ʃ (4 × 14:0, 12:0, 16:0)/UFA (2)
HPI = UFA/Ʃ (4 × 14:0, 12:0, 16:0) (3)
HH = Ʃ (18:1n–9, PUFA)/Ʃ (12:0, 14:0, 16:0), (4)
where UFA is the sum of unsaturated fatty acids, MUFA is the sum of monounsaturated fatty acids, and PUFA is the sum of polyunsaturated fatty acids. 

### 2.6. Statistical Analysis

Statistical analysis was performed and graphs were plotted using R version 4.0.2 [40]. The tidyverse package was used to transform, clear, and visualize data [41]. We employed the principal component analysis to visualize and reduce data dimensionality using two packages: FactorMineR and factorextra [42,43]. Most statistical tests, namely, the Shapiro–Wilk normality test, analysis of variance, Tukey post hoc test, and Kruskal–Wallis test, were performed using the R stats package [40]. The graphs were generated using the ggplot2 package of the tidyverse collection; the multcompView package [44] was used to generate captions for the results of the post hoc analysis in the graphs. The graphs were combined using the ggpubr package [45].

## 3. Results

Fourteen FA, chiefly 18:3n–3, 16:0, and 18:2n–6, were detected in pasture plant samples (Table 3). Saturated FAs (SFAs) were dominated by 16:0, with their percentage in plant leaves increasing by 11.6% after mowing. The content of another SFA, 18:0, also increased in fresh plant shoots after mowing. Total SFAs increased by 19.6% after mowing. By contrast, the content of UFAs decreased after mowing. In lipids of the plants studied, the predominant UFAs were 18:3n–3 and 18:2n–6, and mowing decreased their contents. Conversely, the content of 18:1n–9 and 18:1n–7 was greater in plant shoots after mowing than before mowing. Mowing of plant shoots caused the absolute amount of FA to drop by a factor of two and the content of 18:3n–3 and 18:2n–6 by a factor of three (Table 3). 

More than fifty FAs were found in the Yakutian cattle tissues. Fourteen FAs were found in considerable amounts (>1%). In the muscle tissues, SFAs varied from 31% to 45% (18:0 and 16:0 dominated), MUFAs varied from 18% to 42% (18:1n–9 dominated), and PUFAs varied from 9% to 46% (18:2n–6, 20:4n–6, 22:5n–3, and 20:5n–3 dominated). In the liver, SFAs varied from 41% to 46% (18:0 and 16:0 dominated), MUFAs varied from 13% to 19% (18:1n–9 dominated), and PUFAs varied from 34% to 40% (22:5n–3, 18:2n–6, 20:4n–6, and 20:5n–3 dominated). In the adipose tissue, SFAs varied from 41% to 66% (18:0 and 16:0 dominated), MUFAs varied from 23% to 53% (18:1n–9 dominated), and PUFAs varied from 1% to 2%. Conjugated linoleic acid (CLA) was found in all tissues. In all tissues, except for the tissues of a 7-month-old male, the CLA content did not exceed 0.3%. The 7-month-old male contained 0.5–0.8% of CLA in its muscles, 0.9% in its adipose tissues, and 0.7% in its liver. 

Principal component analysis revealed significant differences in FA compositions between tissues of the Yakutian cattle, which were presented in the two-dimensional space of factors corresponding to the largest eigenvalues that were significant at *p* < 0.05 (Figure 2).

The first component explained 61.4% of the total variance, and it was primarily represented by positive correlations with saturated and monounsaturated FAs (16:0, 14:0, 18:1n–7, 18:1n–5), separating adipose tissue, and negative correlations with PUFAs (20:4n–6, 20:5n–3, 20:3n–6, 22:5n–3, 18:3n–3, 18:2n–6, 20:4n–3, and 22:6n–3), separating muscles and liver. 

The second component explained 20.2% of the total variance, and it was provided by positive correlations with monounsaturated FA (16:1n–7, 18:1n–9), separating all but one (*tensor fasciae latae muscle*) muscles of the 9-year-old female, and negative correlations with 18:0 and 22:6n–3, separating liver and adipose tissue.

Results of the principal component analysis were adjusted using analysis of variance and the Kruskal–Wallis test for normally distributed groups and not normally distributed ones, respectively (Figure 3). 

The amount of SFA contained in the adipose tissue was significantly greater than in the muscles and liver. The muscles and adipose tissue did not differ in their MUFA contents, which were significantly higher than in the liver. PUFA contents were similar in the muscles and liver and were considerably greater than in the fat (Figure 3).

Visceral and subcutaneous fats did not differ in their FA composition. Both adipose tissues had significantly lower levels of 18:2n–6 and 18:3n–3 than the other tissues. Additionally, they did not contain LC-PUFA, namely, 20:3n–6, 20:4n–6, 20:4n–3, 20:5n–3, 22:5n–3, 22:6n–3 (Figure 3). 

The muscles had similar contents of almost all FAs. Only musculus biceps femoris differed from tensor fasciae latae in the content of 16:0 and the level of n–6 PUFA. Musculus biceps femoris contained a higher level of 16:0, but tensor fasciae latae contained a higher level of n–6 PUFA (Figure 3).

The liver contained significantly higher levels of 20:4n–3, 22:5n–3, and 22:6n–3, although 20:4n–3 levels were similar in the liver and tensor fasciae latae; the levels of 16:0, 16:1n–7, 18:1n–9 in the liver were lower compared to those contained in the muscles and adipose tissue. FA levels of the liver were close to those of the muscles and considerably different from the FA levels of the adipose tissue. Only 18:0 and 18:1n–5 contents did not differ significantly between the liver and adipose tissue (Figure 3).

In addition, 16:1n–7 and 18:1n–9 levels tended to be higher in the muscles of the older cattle: 3- and 9-year-old ones (Figure 2 and Figure 3). 

The content of total FA (mg/g of wet weight) in the muscles varied from 5.9 ± 0.3 in the 18-month-old animals to 14.7 ± 3.0 in the 9-year-old one, which differed significantly at *p* < 0.05—Tukey HSD post hoc test. Other significant differences were not found.

To assess the quality of the meat, liver, and fat from the Yakutian cattle as food for humans, we calculated IA, IT, HH, and HPI indexes; n–6/n–3 and PUFA/SFA ratios; and the content of EPA and DHA (mg/g of wet weight) (Table 4).

The lowest IA values were found in the liver and the highest in the fat. The IT values in the liver and muscles were similar to each other and lower by a factor of five compared to the IT values in the adipose tissue. The highest HH and HPI values were found in the liver and the lowest in the fat. The n–6/n–3 ratio in the muscles was significantly higher than in the liver and fat (Table 4). However, in the liver, which contained equal levels of n–3 and n–6 PUFAs, the content of these FAs was high (in sum, about 40% of the total FA), while in the fat, their amount was negligible (Figure 3). The PUFA/SFA ratios in the liver and muscles were similar to each other and higher by a factor of 40 than in the adipose tissue. The EPA + DHA content in the liver was significantly greater than in the muscles, while in the fat, these LC-PUFA were not detected (Table 4). 

## 4. Discussion

The meat and liver of the Yakutian cattle are valuable products for human nutrition. As a rule, farm animals are not considered to be a significant source of physiologically valuable omega–3 PUFAs (EPA and DHA) as the content of these PUFAs in meat is low, ranging between 0.07 and 0.25 mg/g ww [6]. Only the liver may contain substantial amounts of n–3 PUFA, up to 1.1 mg/g ww [21,46]. The amount of EPA + DHA in the meat of the Yakutian cattle was a little higher (0.33 mg/g ww) than in the meat of other agricultural animals, while in the liver of the Yakutian cattle, the EPA + DHA content was higher by a factor of 1.5 compared to the liver of other agricultural animals [46].

The n–6/n–3 and PUFA/SFA ratios are the essential parameters of the nutritional quality of lipids in human food. Dietary requirements of n–6/n–3 and PUFA/SFA ratios recommended by the World Health Organization should be less than 8 and above 0.4, respectively [47]. The n–6/n–3 ratio in the meat and liver of the Yakutian cattle is about 1, which can be regarded as perfect. In some fish species, this ratio is above 1, although aquatic ecosystems are the source of EPA and DHA [6]. The PUFA/SFA ratios in the meat and liver of the Yakutian cattle were lower than in fish but higher than in most meat dairy products and higher by a factor of two than the ratio recommended by the WHO [39,47]. However, in the fat of the Yakutian cattle, the PUFA/SFA ratios were several orders of magnitude lower than in the meat and liver, indicating its low value as food for humans in terms of preventing cardiovascular diseases (CVDs). 

As not all saturated FAs are harmful and not all PUFAs are equally useful, indexes have been developed to assess the effects of fatty acid composition on human health more accurately [38,39]. The indexes of atherogenicity and thrombogenicity characterize the atherogenic and thrombogenic potential of the FA in the food [38]. Unsaturated FAs are considered anti-atherogenic and anti-prothrombic since they inhibit plaque accumulation in the vascular system and reduce the level of cholesterol and clot formation in blood vessels [38,39,48]. The lower the IA and IT values the better the nutritional quality and the lower the risk of developing CVD [39]. The IA values in the meat and liver of the Yakutian cattle are comparable to those in many fish species and some farm animals such as chicken and pigs [49,50,51] and considerably lower than in dairy products [39]. The IT values in the meat and liver of the Yakutian cattle are generally higher than in fish but lower than or comparable to those in meat and dairy products [39]. Therefore, according to IA and IT indexes, the meat and liver of the Yakutian cattle have anti-atherogenic and anti-prothrombic properties.

The HH index was developed by Santos-Silva et al. [52] and modified by Mierliță [53] to determine the effect of the FA composition of food products on the level of plasma cholesterol. The HPI index was proposed by Chen et al. [54]. The higher HH and HPI indexes in food products the more beneficial they are for human health [39]. The HH values in the meat and liver of the Yakutian cattle were considerably higher than in fish and the meat of other farm animals, especially in dairy products [39]. In the adipose tissue, the values of these indexes were lower by a factor of 3–5, and, hence, that tissue was less valuable. Data on HPI indexes in foods are scant, and for meat and liver, they are not available at all. 

Such high-quality indicators of the meat and liver of the Yakutian cattle may be associated with a number of factors: a northern habitat and adaptation to a harsh climate, the diet of the cattle, and the low fat content of the meat.

UFAs have lower melting points than SFAs, and, thus, at low ambient temperatures, high UFA content in the tissues contributes to membrane fluidity and improves the thermogenic function of lipids, as shown for adipocytes [55,56]. The same assumption was made in a study of the FA composition of Yakutian horses, whose tissues were rich in UFAs [21]. 

The Yakutian cattle studied in this work fed on dried grass (seven months out of a year) and on pasture grass (five months out of a year). Feeding pasture grass does not lead to excess fat deposition in the meat of the cattle, and the FA composition of the feed is more favorable [57]. However, even with very high ALA content (30–50%) in the diet of the cattle, the content of this FA in the meat and liver was not high, no more than 5%, and the fat contained no more than 1% of ALA. Interestingly, in the Yakutian horses with a similar diet, the ALA contents in the meat and fat (about 15% and about 20%, respectively) were considerably greater than the ALA contents in the Yakutian cattle tissues [21]. The meat of the cattle fed the diet containing linseed, which is high in ALA, contained no more than 5% of the PUFA [58]. This could suggest that food FAs are transformed to a great degree in the body of the cattle. This suggestion is indirectly supported by the fact that the meat and liver of the cattle contained a diversity of C20 and C22 PUFAs, which were not found at all in the plants that were the staple food for the cattle. The contents of 20:4n–6, 20:5n–3, and 22:5n–3 in the meat of the Yakutian cattle were several-fold greater compared to other animals feeding on grass: bison, cattle, deer, sheep, and Yakutian horses [16,17,18,19,59].

We found a significant effect of mowing on the fatty acid content of the studied plants. It was shown that grass wilting is associated with oxidative losses of PUFAs, mainly ALA, whose proportion of the total FA decreases with a simultaneous increase in the proportion of 16:0 [60]. In our experiments, the level of 16:0 increased, but the level of LA and ALA decreased after mowing, as we observed during 72 h of leaf exposure in natural conditions. In this period, in Yakutia cryolithozone, the weather is dry and very hot. Therefore, in the cut leaves of grasses, there is a loss of water in the cells, and plant cells begin to experience various types of stress (hyperosmotic stress, high air temperature, and drought). Under osmotic stress, the content of unbound water in the cell changes. The inflow and retention of water inside a cell largely depend on the vacuolar membrane [61]. The traumatic damage of shoots (mowing) triggers various regeneration mechanisms in plants, which leads to wound healing and restoration of lost aboveground plant organs through the growth of dormant (axillary) buds of the perennial grasses. Previously, it was shown that new shoots of cereals (oats and bromegrass) grown after mowing and subjected to low-temperature hardening in the permafrost zone of Yakutia had a higher content of LA and ALA in their leaves compared to control plants [21]. Obviously, the processes occurring in plants after mowing lead to a change in their quality for consumers.

The percentages of fatty acids in the muscle tissue of different animals are strongly dependent on the total fat content of the tissue, e.g., [57,62]. Ruminants, like many other animals, primarily contain phospholipid PUFA [57]. At low levels of fat in the muscle, the contribution of phospholipids to total lipids is proportionally greater, and, hence, the percentage of PUFA in such tissues is higher than in the tissues with substantial levels of fat. Conversely, the percentages of MUFA and SFA increase in high-fat tissues [57]. In addition, high-grain diets fed to farm animals lead to an increase in the content of n–6 PUFA in meat, whereas grass-based diets increase the content of n–3 PUFA [17,21,58]. Thus, the grain diet combined with barn feeding produces a double effect: an increase in n–6 PUFA and a higher total fat level of the meat. At the same time, in addition to the diet, the genetic type (breed) and age of the animal also affect fat levels [17,63].

## 5. Conclusions

The meat and liver of the indigenous Yakutian cattle, which feed on grass and live in the harsh climate, are healthy food products that contribute to decreasing the risk of developing CVD because of their rather high content of EPA + DHA, optimal n–6/n–3 and PUFA/SFA ratios, low values of IA and IT indexes, and high values of HH and HPI indexes. Therefore, we believe that this valuable cattle breed should be preserved, and actions should be taken to increase their population while retaining their contemporary housing and feeding conditions. However, the results of this study are based on a small sample size and require further confirmation in experiments with a larger number of animals and controlled conditions.

## Figures and Tables

**Figure 1 foods-12-03226-f001:**
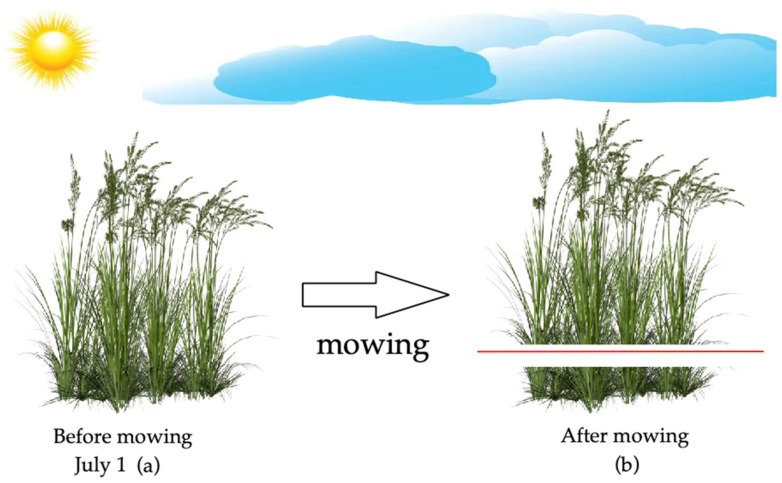
The scheme of a field experiment for evaluating the effect of post-mowing shoots on the FA content of grass leaves.

**Figure 2 foods-12-03226-f002:**
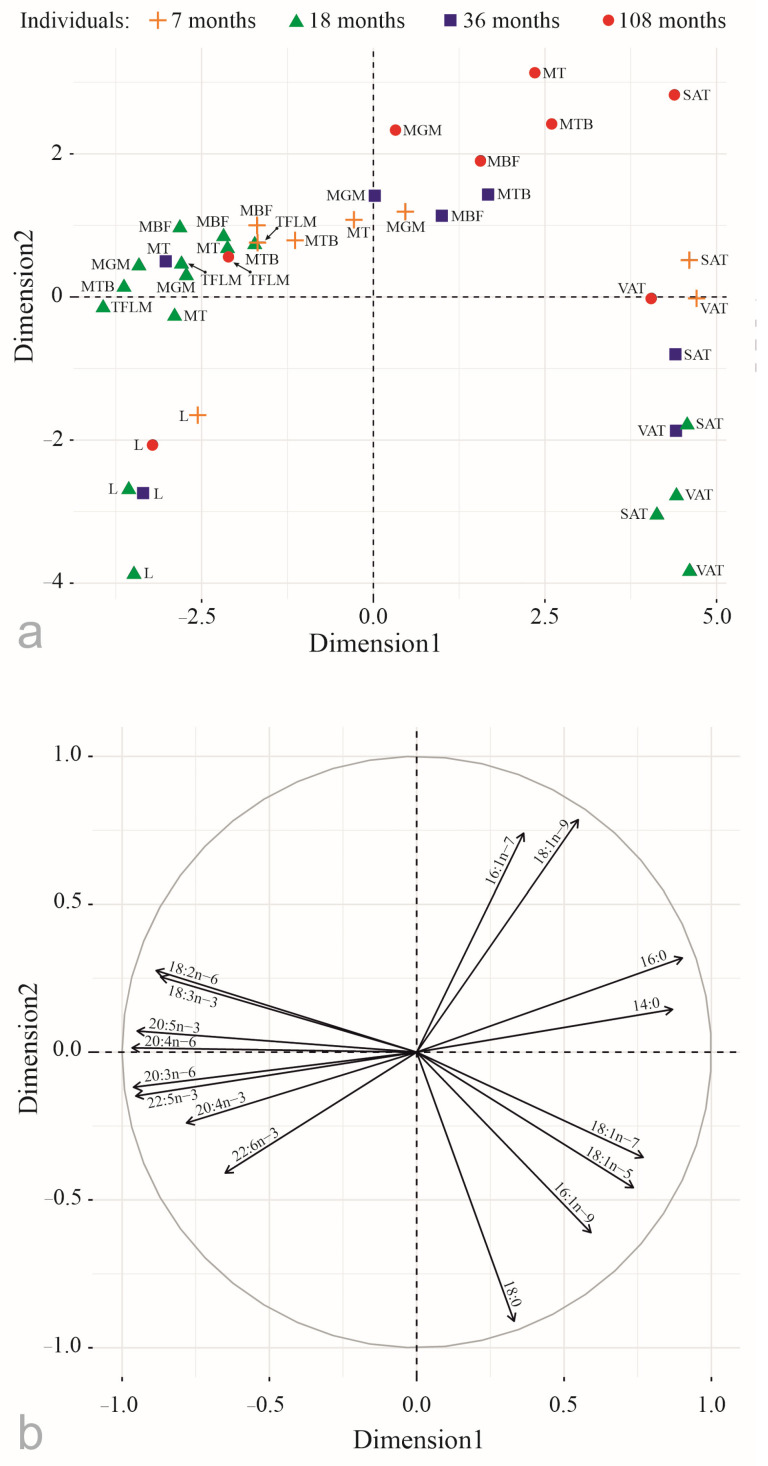
Principal component analysis of percentages of the main fatty acids (**a**) in tissues of the Yakutian cattle (**b**): MT—musculus trapezius; MBF—musculus biceps femoris; MTB—musculus triceps brachii; TFLM—tensor fasciae latae muscle; MGM—musculus gluteus medius; SAT—subcutaneous adipose tissue; VAT—visceral adipose tissue; and L—liver. Dimension 1 and Dimension 2 represent 61.4% and 20.2% of the total variance.

**Figure 3 foods-12-03226-f003:**
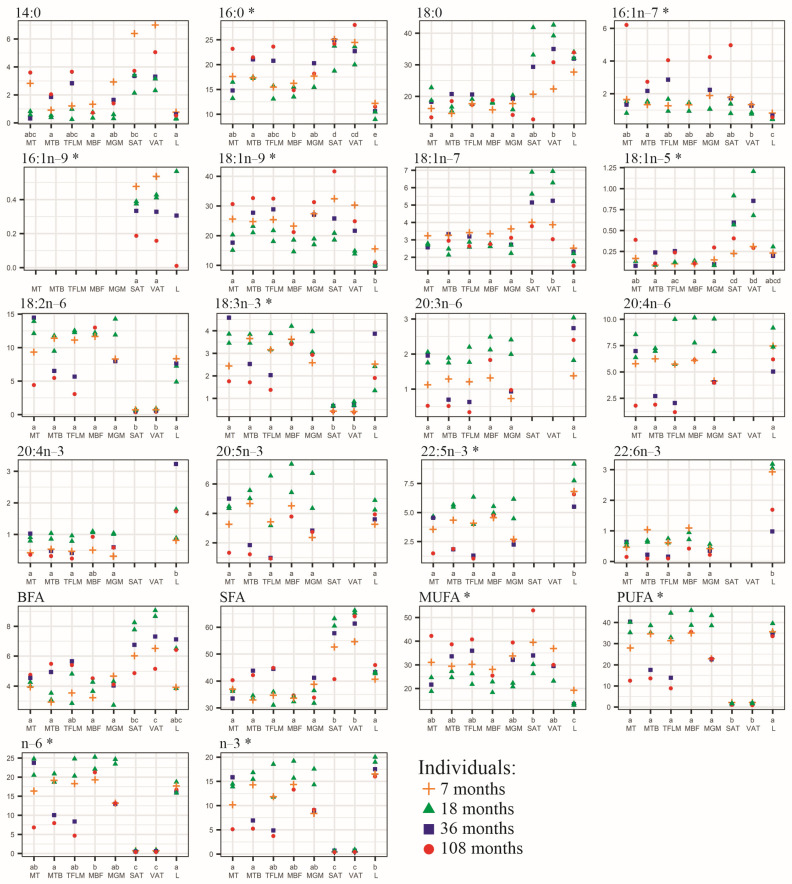
Fatty acid content (% of total FA) in samples of MT—musculus trapezius; MBF—musculus biceps femoris; MTB—musculus triceps brachii; TFLM—tensor fasciae latae muscle; MGM—musculus gluteus medius; SAT—subcutaneous adipose tissue; VAT—visceral adipose tissue; and L—liver. The datasets denoted by the same letters did not differ significantly at *p* < 0.05—Tukey HSD post hoc test (for normally distributed data) and Kruskal–Wallis test (for not normally distributed data, marked by an asterisk * next to the name of the FA). The age of the animals is presented in months. BFA—branched and odd FAs (sum of i13:0, ai13:0, 13:0, i15:0, ai15:0, 15:0, i16:0, i17:0, ai17:0, 17:0, ai17:1, isomers of 17:1); SFA—sum of saturated FAs; MUFA –sum of monounsaturated FAs; PUFA—sum of polyunsaturated FAs.

**Table 1 foods-12-03226-t001:** Species composition of plants and their stages of development.

№	Species	Family	Stages of Development (July 1)	Sample
1	*Calamagrostis neglecta* (Ehrh.) *Gaertn., B. Mey. & Schreb.*	Poaceae	Stem elongation	leaves
2	*Calamagrostis langsdorffii* (Link) Trin.	Poaceae	Tillering	leaves
3	*Arctophila fulva* (Trin.) Andersson	Poaceae	Stem elongation	leaves
4	*Glyceria aquatica* (L.) Wahlb., nom. illeg.	Poaceae	Ear emergence	leaves
5	*Beckmannia syzigachne* (Steud.) Fernald	Poaceae	Tillering	leaves

**Table 2 foods-12-03226-t002:** Meteorological parameters during the research period (Batagay-Alyta, Eveno-Bytantaysky District, the Republic of Sakha (Yakutia), Russia).

Sampling Date	Daily Average Air Temperature, °C *	Total Precipitation, mm **	Photoperiod, h
1 July (Before mowing)	19.2 ± 5.5	9.0	21.0
2 July	21.7 ± 9.0	7.6	20.1
3 July	24.3 ± 7.9	5.0	20.1
4 July (After mowing 72 h)	20.2 ± 6.7	0.5	20.0

* Average data for 24 h prior to sampling (±SEM); ** the amount within 10 days before sampling. Data were taken from the sites http://www.pogodaiklimat.ru/weather.php?id=24261 (accessed on 10 June 2023).

**Table 3 foods-12-03226-t003:** The content of fatty acids (mg/g of dry weight ± SEM; % of total FA ± SEM) in pasture plants before and after mowing (Eveno-Bytantaysky District, Yakutia).

	Before Mowing, n = 4		After Mowing, n = 4	
FA	mg/g DW	% of Total FA	mg/g DW	% of Total FA
12:0	trace	0.5 ± 0.0 ^B^	0.1 ± 0.0	1.2 ± 0.2 ^A^
14:0	trace	0.8 ± 0.2 ^B^	0.1 ± 0.1	2.7 ± 0.1 ^A^
15:0	trace	0.3 ± 0.1 ^A^	trace	0.7 ± 0.2 ^A^
16:0	2.4 ± 0.0 ^a^	22.5 ± 1.4 ^B^	1.8 ± 0.7 ^a^	34.1 ± 0.6 ^A^
16:1n–9	0.1 ± 0.0 ^a^	0.6 ± 0.2 ^B^	0.1 ± 0.0 ^a^	1.8 ± 0.1 ^A^
16:1n–5	0.3 ± 0.0 ^a^	2.5 ± 0.0 ^A^	0.1 ± 0.0 ^b^	1.0 ± 0.2 ^B^
17:0	trace	0.2 ± 0.0 ^B^	trace	0.7 ± 0.1 ^A^
18:0	0.2 ± 0.0 ^a^	1.9 ± 0.2 ^B^	0.3 ± 0.2 ^a^	5.7 ± 0.8 ^A^
18:1n–9	0.2 ± 0.0 ^a^	2.2 ± 0.2 ^B^	0.2 ± 0.1 ^a^	4.4 ± 0.5 ^A^
18:1n–7	0.1 ± 0.0 ^a^	0.8 ± 0.0 ^B^	0.1 ± 0.1 ^a^	2.4 ± 0.5 ^A^
18:2n–6	1.1 ± 0.0 ^a^	10.0 ± 0.1 ^A^	0.4 ± 0.1 ^b^	8.1 ± 1.1 ^B^
18:3n–3	5.8 ± 0.5 ^a^	54.9 ± 1.9 ^A^	1.8 ± 0.6 ^a^	33.5 ± 1.3 ^B^
20:0	0.1 ± 0.0 ^a^	1.1 ± 0.2 ^A^	0.1 ± 0.0 ^a^	1.7 ± 0.3 ^A^
22:0	0.2 ± 0.0 ^a^	1.8 ± 0.4 ^A^	0.1 ± 0.0 ^a^	1.9 ± 0.3 ^A^
SFA	3.1 ± 0.1 ^a^	29.1 ± 2.2 ^B^	2.6 ± 1.1 ^a^	48.7 ± 1.3 ^A^
UFA	7.5 ± 0.7 ^a^	70.9 ± 2.2 ^A^	2.7 ± 1.0 ^b^	51.3 ± 1.3 ^B^
Total FA	10.6 ± 0.6 ^a^		5.3 ± 2.1 ^b^	

Different letters indicate significant differences (*p* ≤ 0.05) between FA content (mg/g DW—lowercase letter—and % of total FA—uppercase letter) in pasture plants before and after mowing after performing Student′s *t*-test.

**Table 4 foods-12-03226-t004:** The mean values (±SEM) of the index of atherogenicity (IA), index of thrombogenicity (IT), and health-promoting index (HPI); hypocholesterolemic/hypercholesterolemic (HH), n–6/n–3, PUFA/SFA, and the content of EPA and DHA (mg/g of wet weight) in the meat, liver, and fat of the Yakutian cattle.

Parameters	Meat	Liver	Fat
IA *	0.39 ± 0.03 ^B^	0.25 ±0.02 ^C^	1.17 ± 0.08 ^A^
IT *	0.65 ± 0.05 ^B^	0.62 ± 0.03 ^B^	3.13 ±0.28 ^A^
HH	3.05 ± 0.18 ^B^	4.25 ± 0.25 ^A^	0.94 ±0.08 ^C^
HPI	2.91 ± 0.19 ^B^	4.07 ± 0.29 ^A^	0.89 ± 0.07 ^C^
n–6/n–3	1.47 ± 0.03 ^A^	0.96 ± 0.04 ^B^	1.09 ± 0.10 ^B^
PUFA/SFA	0.80 ± 0.07 ^A^	0.75 ± 0.04 ^A^	0.02 ± 0.002 ^B^
EPA + DHA, mg/g WW	0.33 ± 0.01 ^B^	1.59 ± 0.12 ^A^	0.00 ± 0.00 ^C^

The datasets denoted by the same letters did not differ significantly at *p* < 0.05—Tukey HSD post hoc test (for normally distributed data) and Kruskal–Wallis test (for not normally distributed data, marked by an asterisk *).

## Data Availability

The data used to support the findings of this study can be made available by the corresponding author upon request.

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
