# Peer review of "Preliminary Estimation of Nutritional Quality of the Meat, Liver, and Fat of the Indigenous Yakutian Cattle Based on Their Fatty Acid Profiles"

_foods, 2023, doi:10.3390/foods12173226_

Round 1

Reviewer 1 Report

The manuscript “Nutritional Quality of Meat, Liver, and Fat of the Indigenous Yakutian Cattle Based on their Fatty Acid Profiles” examines the fatty acid composition of different tissues collected from rare Yakutian cattle and the fatty acid composition of pasture plants before and after mowing. The study may have some potential  but I have concerns regarding the overall design of the experiment and consequently the result interpretation.

ï‚· My main concern is with the sample size and diversity. Only 5 animals were included in the experiment (4 males and 1 female) considerably differing in their age (one 7-month, two 18-month, one 3-year, one 9-year old). As a result, data were often not normally distributed. A low group size may result in a reduced test power and an increased probability of Type II error occurrence and may therefore influence the interpretation of the results obtained. It is also associated with a high variability of FA proportions especially in muscles and adipose tissues. I am aware that at the time of the slaughter there may not have been more animals available, as Yakutian is a breed that is not probably very widespread. Nevertheless, the low sample size and uniformity may significantly affect the results of the study. The results should therefore be interpreted with some caution, and the fact of the low sample size should be noted in the text. The age categories are indicated as “groups” in Figure 2 and Figure 3 which is not true because they are mostly just individuals.

There are statements throughout the manuscript that describe the effect of animal age on FA composition (e.g. L296-300, L402-404). However, this effect was not (could not be) statistically tested and therefore these statements should be removed.

Other comments

ï‚· Table 2: What is the measure of variability used in Table 2? Standard deviation?

ï‚· L167: From where the subcutaneous and visceral samples were taken?

ï‚· Table 3: How many samples were analysed? Suggested: Line 1 of the table: Before mowing (n = ?); After mowing (n = ?)

ï‚· L321 and L326: Two WHO recommendations on FA ratios are cited. The newest recommendation should be used.

ï‚· L392: I agree that muscle FA composition is strongly dependent of intramuscular fat content. Was it measured in this study? It may have explained some differences observed.

Author Response

Authors: We would like to thank the esteemed Reviewers for the interest in our work and careful analysis of the results.

Response to Reviewer 1

The manuscript “Nutritional Quality of Meat, Liver, and Fat of the Indigenous Yakutian Cattle Based on their Fatty Acid Profiles” examines the fatty acid composition of different tissues collected from rare Yakutian cattle and the fatty acid composition of pasture plants before and after mowing. The study may have some potential  but I have concerns regarding the overall design of the experiment and consequently the result interpretation.

My main concern is with the sample size and diversity. Only 5 animals were included in the experiment (4 males and 1 female) considerably differing in their age (one 7-month, two 18-month, one 3-year, one 9-year old). As a result, data were often not normally distributed. A low group size may result in a reduced test power and an increased probability of Type II error occurrence and may therefore influence the interpretation of the results obtained. It is also associated with a high variability of FA proportions especially in muscles and adipose tissues. I am aware that at the time of the slaughter there may not have been more animals available, as Yakutian is a breed that is not probably very widespread. Nevertheless, the low sample size and uniformity may significantly affect the results of the study. The results should therefore be interpreted with some caution, and the fact of the low sample size should be noted in the text. The age categories are indicated as “groups” in Figure 2 and Figure 3 which is not true because they are mostly just individuals.

Authors: The Reviewer is absolutely right that the number of animals was limited by the small number of this breed and the remoteness of the farms. However, we used animals of different ages, which made it possible to obtain more complete data on the nutritional value of the breed. In addition, in confirmation that even this small number of samples characterizes the breed, it is indicated that different muscles of the same individual had a high variability, as shown in Fig. 2. We've color-coded Figures 2 and 3 to help readers better see the variability in muscle FA composition.

We have added the number of animals in Abstract.

In figures 2 and 3, we have replaced "groups" with "individuals".

There are statements throughout the manuscript that describe the effect of animal age on FA composition (e.g. L296-300, L402-404). However, this effect was not (could not be) statistically tested and therefore these statements should be removed.

Authors: Done.

Table 2: What is the measure of variability used in Table 2? Standard deviation?

Authors: We used the standard error of the mean (± SEM). We have added this information.

L167: From where the subcutaneous and visceral samples were taken?

Authors: We have added this information.

Table 3: How many samples were analysed? Suggested: Line 1 of the table: Before mowing (n = ?); After mowing (n = ?)

Authors: Done.

L321 and L326: Two WHO recommendations on FA ratios are cited. The newest recommendation should be used.

Authors: We used only one WHO recommendation. The reference on page L326 was to a mini-review with food data.

L392: I agree that muscle FA composition is strongly dependent of intramuscular fat content. Was it measured in this study? It may have explained some differences observed.

Authors: We measured total FA content (mg/g WW) that directly relates to fat content. We have made the comparison between animals (sum in all muscles) and added this data to Results. The 9-year-old animal had in its muscles significantly higher fat content than the other individuals.

Reviewer 2 Report

Some adjustments needed to improve readability and comprehension.

-       Respect the journal's instructions for references.

-       Correct typos.

-       Correct ml to mL

-       It is not clear in Table 3 between what the comparisons are. They should be inside before and after the mowing per unit of measurement. So, it is not clear. It is also customary to use the first letter "A" for the biggest value. To review.

-       Better describe the two graphics what they represent in the caption of Figure 2.

-       In table 4 use the first letter "A" for the biggest value. To review.

-       Lines 233, 252. In point 2.6 you indicated the analysis of the principal components while in these lines are reported “Canonical correspondence analysis”. Clarify.

-       Lines 244 and 248. Better “The first component” then “Factor 1”.

-       Line 286. Table 4 must be placed after.

-       Line 297, 300. Use english version, no “vs.” but “vs”.

-       Lines 303-304. Not clear

-       Lines 307-309. Are these lines part of Figure 4? It is not clear.

-       Line 344. Indicate the first author.

Author Response

Authors: We would like to thank the esteemed Reviewers for the interest in our work and careful analysis of the results.

Response to Reviewer 2

-       Respect the journal's instructions for references.

Authors: Done.

-       Correct typos.

Authors: Done.

-       Correct ml to mL

Authors: Done.

-       It is not clear in Table 3 between what the comparisons are. They should be inside before and after the mowing per unit of measurement. So, it is not clear. It is also customary to use the first letter "A" for the biggest value. To review.

Authors: Done.

-       Better describe the two graphics what they represent in the caption of Figure 2.

Authors: Done.

-       In table 4 use the first letter "A" for the biggest value. To review.

Authors: Done.

-       Lines 233, 252. In point 2.6 you indicated the analysis of the principal components while in these lines are reported “Canonical correspondence analysis”. Clarify.

Authors: Done.

-       Lines 244 and 248. Better “The first component” then “Factor 1”.

Authors: Done.

-       Line 286. Table 4 must be placed after.

Authors: Done.

-       Line 297, 300. Use english version, no “vs.” but “vs”.

Authors: According to recommendation of Reviewer 1 it was deleted.

-       Lines 303-304. Not clear

Authors: We have added explanations.

-       Lines 307-309. Are these lines part of Figure 4? It is not clear.

Authors: Yes, these lines are part of Figure 3.

-       Line 344. Indicate the first author.

Authors: We have added the names of the first authors:

The HH index was developed by Santos-Silva et al. [52] and modified by Mierliță [53] to determine the effect of FA composition of food products on the level of plasma cholesterol. The HPI index was proposed by Chen et al. [54].

Reviewer 3 Report

The manuscript submitted by Makhutova et al. evaluates the impact of grass feeding on the fatty acid profile in different cuts of meat, liver, and fat from Yakutian cattle raised in cold climates in the Yakutia area. Currently, there are only a few thousand specimens of this autochthonous breed, for which the interest of the authors is to evaluate if the typical feeding method in the area, together with the cold climate, has an impact on producing meat with healthier characteristics for the consumer. From the scientific and social point of view, finding food sources with a healthier profile for the consumer is essential. However, as this research is proposed, it has significant limitations for its publication. 

The deposition of fatty acids in the animal depends on several factors, among which stand out whether the animal is a ruminant, type of diet, animal gender, climatic conditions where the animals are, animal age, and even the breed of cattle also influences the deposition of fat. Because many factors influence fat deposition and lipid profile, the realization of the current investigation to analyze whether feeding with grass under cold weather generates favorable changes for health in the lipid profile of Yakutian cattle meat should have been more extensive in the number and uniformity of animals to be evaluated.

Some observations of the manuscript are described below:

L 165-169. Concerning the sentence, “Samples of muscles (trapezius, triceps brachii, tensor fascia latae, biceps femoris, gluteus medius), subcutaneous fat, visceral fat, and liver were harvested from the cattle of different ages. Those were four males – a 7-month-old one, two 18-month-old ones, and a 3- year- old one – and a 9-year-old female”.  

The number of animals used to conduct this research needs to be improved to determine the effect of grass feeding on the lipid profile of deposited fat. In addition to the fact that it needs to be an adequate number of animals, not using homogeneous animals is incorrect since age is one factor that influences the fat deposit. The authors also used an adult cow, which affects the lipid profile by gender and comparative age of the animal.

L 26-27. Regarding the sentence, “The results of the present study support the importance of preserving this valuable cattle breed. Actions should be taken to increase their population while retaining their contemporary housing and feeding conditions”

As previously mentioned, the minimum number of animals used in the research, as well as the difference in age and gender of the animals, does not allow us to argue the assumption mentioned in the lines above.

L 84-90. Respecto to the sentence, “Since 1929, the Yakutian cattle have been crossed with the Simmental and the Kholmogory cattle breeds to increase their productivity. The genetic study conducted in 2005 compared two hybrid cattle populations [30]. The study showed that the genetic contribution of the indigenous Yakutian cattle to the Yakutian-Kholmogory population was low whereas substantial genetic contribution of the Yakutian cattle was revealed in the genome of the Yakutian-Simmental population. Purebred Yakutian cattle can still be found in remote northern family farms in Yakutia”

As a suggestion to improve the work in the future, the authors could use more purebred Yakutian cattle animals and make comparisons with Yakutian cattle crossed with Simmental and Kholmogory to determine if it is an effect of the Yakutian breed and not of the feeding system or cold weather.

L 395-401. Regarding the sentences, “Meat and liver of the indigenous Yakutian cattle, which feed on grass and live in the harsh climate, are health food products that contribute to decreasing the risk of developing CVD because of their rather high content of EPA + DHA, optimal n-6/n-3 and PUFA/SFA ratios, low values of IA and IT indexes, and high values of HH and HPI indexes. Therefore, we believe that this valuable cattle breed should be preserved, and actions should be taken to increase their population while retaining their contemporary housing and feeding conditions”

According to how this research was conducted (number of animals and different ages and gender), there are no arguments to conclude the work in this way.

Author Response

Authors: We would like to thank the esteemed Reviewers for the interest in our work and careful analysis of the results.

Response to Reviewer 3

The manuscript submitted by Makhutova et al. evaluates the impact of grass feeding on the fatty acid profile in different cuts of meat, liver, and fat from Yakutian cattle raised in cold climates in the Yakutia area. Currently, there are only a few thousand specimens of this autochthonous breed, for which the interest of the authors is to evaluate if the typical feeding method in the area, together with the cold climate, has an impact on producing meat with healthier characteristics for the consumer. From the scientific and social point of view, finding food sources with a healthier profile for the consumer is essential. However, as this research is proposed, it has significant limitations for its publication.

Authors: The purpose of the present study was to estimate nutritional value of the meat of Yakutian cattle for humans. As the FA composition of the food has a strong effect on the FA composition of the animals consuming it, we presented the composition of the pasture plants on which all study animals had been feeding. Humans consume meat of the cattle of different ages, and, thus, a study of animals of different ages offers a fuller picture of the value of these animals as a source of food for humans. We were not able to use a larger number of animals in this study. The number of animals was limited by their availability (farmers are not willing to cooperate with strangers) and the remoteness of farms. In addition, in confirmation that even this small number of samples characterizes the breed, it is indicated that different muscles of the same individual had a high variability, as shown in Fig. 2. We've color-coded Figures 2 and 3 to help readers better see the variability in muscle FA composition.

The deposition of fatty acids in the animal depends on several factors, among which stand out whether the animal is a ruminant, type of diet, animal gender, climatic conditions where the animals are, animal age, and even the breed of cattle also influences the deposition of fat. Because many factors influence fat deposition and lipid profile, the realization of the current investigation to analyze whether feeding with grass under cold weather generates favorable changes for health in the lipid profile of Yakutian cattle meat should have been more extensive in the number and uniformity of animals to be evaluated.

Authors: That was not our task. This is the first study to characterize the nutritional value of this breed of cattle in terms of fatty acid composition. 

Some observations of the manuscript are described below:

L 165-169. Concerning the sentence, “Samples of muscles (trapezius, triceps brachii, tensor fascia latae, biceps femoris, gluteus medius), subcutaneous fat, visceral fat, and liver were harvested from the cattle of different ages. Those were four males – a 7-month-old one, two 18-month-old ones, and a 3- year- old one – and a 9-year-old female”. 

The number of animals used to conduct this research needs to be improved to determine the effect of grass feeding on the lipid profile of deposited fat. In addition to the fact that it needs to be an adequate number of animals, not using homogeneous animals is incorrect since age is one factor that influences the fat deposit. The authors also used an adult cow, which affects the lipid profile by gender and comparative age of the animal.

Authors: In the present study, we estimated the nutritional value of the Yakutian cattle and showed that their FA composition differs considerably from the FA composition of their food, i.e. the level of transformation of the consumed fatty acids is very high. 

L 26-27. Regarding the sentence, “The results of the present study support the importance of preserving this valuable cattle breed. Actions should be taken to increase their population while retaining their contemporary housing and feeding conditions”

As previously mentioned, the minimum number of animals used in the research, as well as the difference in age and gender of the animals, does not allow us to argue the assumption mentioned in the lines above.

Authors: In our study, we process the samples statistically, and all conclusions are based on statistically significant differences. A change in the number of the samples changes the error of the mean and significance of the differences. There is no reason why the 5 study individuals should differ in their FA composition from the other animals of this population, which are raised under the same conditions.

L 84-90. Respecto to the sentence, “Since 1929, the Yakutian cattle have been crossed with the Simmental and the Kholmogory cattle breeds to increase their productivity. The genetic study conducted in 2005 compared two hybrid cattle populations [30]. The study showed that the genetic contribution of the indigenous Yakutian cattle to the Yakutian-Kholmogory population was low whereas substantial genetic contribution of the Yakutian cattle was revealed in the genome of the Yakutian-Simmental population. Purebred Yakutian cattle can still be found in remote northern family farms in Yakutia”

As a suggestion to improve the work in the future, the authors could use more purebred Yakutian cattle animals and make comparisons with Yakutian cattle crossed with Simmental and Kholmogory to determine if it is an effect of the Yakutian breed and not of the feeding system or cold weather.

Authors: Thank you for your suggestion. We intend to continue our study if we have an opportunity to collect new data.

L 395-401. Regarding the sentences, “Meat and liver of the indigenous Yakutian cattle, which feed on grass and live in the harsh climate, are health food products that contribute to decreasing the risk of developing CVD because of their rather high content of EPA + DHA, optimal n-6/n-3 and PUFA/SFA ratios, low values of IA and IT indexes, and high values of HH and HPI indexes. Therefore, we believe that this valuable cattle breed should be preserved, and actions should be taken to increase their population while retaining their contemporary housing and feeding conditions”

According to how this research was conducted (number of animals and different ages and gender), there are no arguments to conclude the work in this way.

Authors: There are no such limitations based on the statistical analysis we used.

Reviewer 4 Report

 I believe that this valuable cattle breed should be preserved and know for scientific community. But, the number of animals is too small to draw conclusions. There are one group have 1 animal. Statistical analysis is not very robust. The numbers of animals of per experimental groups is clearly insufficient. Fatty acid composition of meat, liver and fat is affected diet, species, and other factors.  The authors must not be compared their results with other species.

Good.  

Author Response

Authors: We would like to thank the esteemed Reviewers for the interest in our work and careful analysis of the results.

Response to Reviewer 4

I believe that this valuable cattle breed should be preserved and know for scientific community. But, the number of animals is too small to draw conclusions. There are one group have 1 animal. Statistical analysis is not very robust. The numbers of animals of per experimental groups is clearly insufficient. Fatty acid composition of meat, liver and fat is affected diet, species, and other factors.  The authors must not be compared their results with other species.

Authors: We agree with the Reviewer that the number of animals is small. We were not able to use a larger number of animals in this study. The number of animals was limited by their availability (farmers are not willing to cooperate with strangers) and the remoteness of farms. In addition, in confirmation that even this small number of samples characterizes the breed, it is indicated that different muscles of the same individual had a high variability, as shown in Fig. 2. We've color-coded Figures 2 and 3 to help readers better see the variability in muscle FA composition.

The only thing that we can do in this stage is to change the title of the manuscript to Preliminary Estimation of Nutritional Quality of Meat, Liver, and Fat of the Indigenous Yakutian Cattle Based on Their Fatty Acid Profiles

Round 2

Reviewer 1 Report

Comments:

I am still worried about the sample size. In my opinion the breed cannot be satisfactorily characterized on the basis of data from 5 animals of different sex and age reared in poorly defined conditions.

L169: Please replace thigh with rump

The low number of animals represents a limitation of the study and therefore, in conclusion, if the manuscript is accepted for publication, it should be stressed that the results of the study should be taken with caution due to the small sample size, and that experiments with more animals and better controlled conditions are needed to confirm them.

Author Response

I am still worried about the sample size. In my opinion the breed cannot be satisfactorily characterized on the basis of data from 5 animals of different sex and age reared in poorly defined conditions.

L169: Please replace thigh with rump

Authors: Done.

The low number of animals represents a limitation of the study and therefore, in conclusion, if the manuscript is accepted for publication, it should be stressed that the results of the study should be taken with caution due to the small sample size, and that experiments with more animals and better controlled conditions are needed to confirm them.

Authors: We added in Conclusion “However, the results of the study are based on the small sample size and require further confirmation in experiments with a larger number of animals and controlled conditions.”

Reviewer 3 Report

This reviewer considers that the authors' responses are appropriate and support the questions raised during the first round. Therefore, I agree with the content of the article in its current form and, in my opinion, it should be taken into account for publication.

Author Response

The team of authors expresses their deep gratitude to the Reviewer for a deep professional analysis of our article and for valuable and useful comments.